# Domain-Specific On-Device Object Detection Method

**DOI:** 10.3390/e24010077

**Published:** 2022-01-01

**Authors:** Seongju Kang, Jaegi Hwang, Kwangsue Chung

**Affiliations:** Department of Electronics and Communications Engineering, Kwangwoon University, Seoul 01897, Korea; sjkang@cclab.kw.ac.kr (S.K.); jghwang@cclab.kw.ac.kr (J.H.)

**Keywords:** object detection, domain-specific, on-device, lightweight network

## Abstract

Object detection is a significant activity in computer vision, and various approaches have been proposed to detect varied objects using deep neural networks (DNNs). However, because DNNs are computation-intensive, it is difficult to apply them to resource-constrained devices. Here, we propose an on-device object detection method using domain-specific models. In the proposed method, we define object of interest (OOI) groups that contain objects with a high frequency of appearance in specific domains. Compared with the existing DNN model, the layers of the domain-specific models are shallower and narrower, reducing the number of trainable parameters; thus, speeding up the object detection. To ensure a lightweight network design, we combine various network structures to obtain the best-performing lightweight detection model. The experimental results reveal that the size of the proposed lightweight model is 21.7 MB, which is 91.35% and 36.98% smaller than those of YOLOv3-SPP and Tiny-YOLO, respectively. The f-measure achieved on the MS COCO 2017 dataset were 18.3%, 11.9% and 20.3% higher than those of YOLOv3-SPP, Tiny-YOLO and YOLO-Nano, respectively. The results demonstrated that the lightweight model achieved higher efficiency and better performance on non-GPU devices, such as mobile devices and embedded boards, than conventional models.

## 1. Introduction

Image recognition is a significant activity in the field of computer vision [1]. Earlier, image recognition was used for simple tasks, such as face or character recognition [2,3]. However, with the advent of solutions to detect more varied objects using neural networks, research into object detection has become increasingly active [4,5,6]. Object detection is more complex than image recognition as it involves finding objects in digital images and predicting the object classes to which they belong. As the performance capabilities of available hardware have improved in recent years, and as various open datasets for object detection have been released, it is possible to train a deep neural network (DNN) model with a complex structure for various domains.

When applied for object detection, a DNN model has hundreds of hidden layers and tens of millions of parameters that perform localization and classification tasks. Further, as DNN models become more complex for highly accurate object detection, the computational capabilities of the device performing the object recognition have become increasingly important. To enable object detection in resource-constrained environments, many researchers have been studying techniques such as model compression and data offloading. However, it is still challenging to satisfy the accuracy and latency requirements of applications in resource-constrained environments because of accuracy degradation resulting from model compression and the instability of data offloading methods depending on bandwidth conditions.

Therefore, a new paradigm is needed for object detection in resource-constrained environments. To satisfy the latency requirements of object detection, the runtime should be affected exclusively by the complexity of the detection model, and no other factors such as bandwidth. In addition, different detection models should be used for each domain by defining the related objects for each domain to satisfy the accuracy requirements. Because existing object detection models use information that have low relevance to the domain, computational resources are wasted and there is significant computational overhead. However, if the domain in which object detection is to be performed is known beforehand, we can define object classes that are related to that domain. To solve this problem, we apply a divide-and-conquer approach, which solves a large problem in the general domain by dividing it into smaller, domain-specific problems.

Here, we proposed a domain-specific on-device object-detection method. We defined objects with a high frequency of appearance as the object of interest (OOI) group and trained different detection models for each OOI group. Because the OOI group for each domain contains a small number of classes, the detection model detects objects using a shallow network structure. The proposed method ensures that the latency requirement is met by reducing the complexity of the detection models; also, it ensures the accuracy requirement as object detection is performed only in OOI groups. This study’s main contributions are:A method for defining the OOI group for each domain showing how to generate training datasets for each OOI group. To solve the problems of class imbalance, we designed a method to collect data using a state-of-the-art (SOTA) DNN model.Many structures that can improve the performance of lightweight networks are presented. Additionally, we designed object detection models that are suitable for on-device environments, and through experiments we proposed the best-performing detection model.

The remainder of this paper is organized as follows. Section 2 discusses previous studies that have applied DNNs in resource-constrained environments. In addition, we present several elements to consider for lightweight networks and describe existing efficient lightweight networks. Section 3 proposes a domain-specific on-device object detection method. Section 4 evaluates the performance of the proposed lightweight model using open datasets. Finally, Section 5 concludes the paper.

## 2. Materials and Methods

### 2.1. Deep Neural Network

After AlexNet won the ILSVRC 2012 challenge with its outstanding performance, DNNs began to garner considerable research attention, and various network architectures have since emerged, including VGGNet (2014), GoogleNet (2015), and ResNet (2015) [4,5,6]. DNN models are being designed to be deeper and more complex, and pre-training and fine-tuning methods using large image datasets have become mainstream. Because DNN models have hundreds of layers and millions of parameters, the inference machine must perform computation-intensive workloads. The runtime of DNN models can be reduced by using accelerators such as GPUs, NPUs, and TPUs. With the development of these hardware devices, DNNs have been applied in various fields, such as object detection, image recognition, and speech recognition.

### 2.2. Previous Research in Resource-Constrained Environment

With the growing demand for deep learning-based intelligent applications, many studies have been conducted to design lightweight models that can operate in resource-constrained environments, such as mobile devices and embedded boards. To reduce the number of parameters of DNN models, Bucilua et al. [7] proposed model compression, which approximates the function learned by a complex model into a much smaller, faster model with comparable performance. With a fast and resource-efficient model, albeit slightly less accurate, applications that need to meet low latency requirements can perform better than with an uncompressed model. Wu et al. [8] proposed a unified framework called Quantized CNN that simultaneously accelerates and compresses convolutional neural networks (CNNs). Quantized CNN achieves outstanding speed-up and compression rates, with only negligible loss of classification accuracy, through parameter quantization. Rastegari et al. [9] proposed XNOR-Net to reduce the runtime of DNNs and the size of the network. XNOR-Net approximates all weight values to binary and uses bitwise operations. Han et al. [10] proposed weight pruning to reduce the number of floating-point operations. Specifically, it reduces the required computational resources by converting the weight of small values to zero based on the premise that fewer weighted values have a lower impact on inferences. Model compression, parameter quantization, and weight pruning enable inferences within a short latency period. However, model compression and weight pruning can lead to poor performance in accuracy-critical applications because they incur accuracy degradation of the model. In addition, although Quantized CNN [8] and XNOR-Net [9] have improved classification accuracy, they have not been evaluated for object detection performance.

Many researchers have also investigated data offloading, in which data are transferred to central servers or clouds without reducing the complexity of the model [11,12]. The main idea of data offloading is to transfer data to a central server that has sufficient computing resources to perform the DNN operations. Because the DNN model has many computationally intensive workloads, mobile devices must use the computing resources of edge nodes or cloud servers. If a device offloads all of its workloads to a server, the performance of the DNN will depend exclusively on the network bandwidth and the amount of data transmitted. In addition, studies have been conducted to selectively transfer workloads or to migrate workloads through cross-node collaboration for more efficient data offloading [13,14]. However, unstable bandwidth can lead to performance degradation due to the latency experienced during data upload or download.

Marco et al. [15] proposed a model selection approach that selects the optimal model to analyze input data with high accuracy. Figure 1 shows an example of the model selection. Training data with similar patterns are clustered into the same group. Each group is then paired with a DNN model that can achieve optimal performance. If there is more than one optimal model for the feature pattern, the model with the shortest runtime is defined as the optimal model. However, when the optimal model is the most complex DNN model, such as ResNet, unnecessary overhead results owing to the model selection process.

### 2.3. Important Elements for Lightweight Object Detection

Object detection performance depends on various factors, such as the structure of the network, the quality of the training data, and the number of classes to be detected. In this section, we discuss in more detail the factors that affect the performance of object detection.

#### 2.3.1. Backbone Network

In the backbone network, there are many training parameters for feature extraction; high-level features are extracted from low-level features. Therefore, the greater the number of hidden layers in the backbone, the more accurate feature map extraction can be. After AlexNet won the ILSVRC 2012, various DNN models with dozens of convolution layers, including the VGGNet-16 model, emerged. However, if the number of hidden layers is increased excessively, the training parameters of the front layers will converge faster than those of the back layers, resulting in the vanishing gradient problem. To address the vanishing gradient problem, He et al. [6] proposed a residual network (ResNet) structure that adds a shortcut to the network. The main concept of ResNet is that when identity mapping is optimal, it simply zeros out the residuals using a stack of nonlinear layers. Because the addition operation is performed without an additional layer, the complexity of the model is maintained. However, because ResNet performs an addition operation, the uniqueness of the feature map cannot be maintained. To solve this problem, He et al. [16] proposed the densely connected convolutional network (DenseNet), wherein each layer accepts feature mappings from all previous layers. DenseNet reuses the unique feature map of the previous layer through the concatenate operation. This allows the network to be more simplified and addresses the vanishing gradient problem with fewer learning parameters than ResNet.

#### 2.3.2. Quality of the Dataset

Datasets are key factors in determining the performance of a model. Overfitting or underfitting problems can occur as a result of class imbalance problems. In addition, the performance of the model is affected by factors such as the reliability of the dataset, size of the objects, and number of objects per image. Although reliable datasets such as MS COCO, Visdrone, and Pascal VOC are available, it is difficult to find a dataset that completely satisfies the purpose of the model [17,18,19]. In some cases, the desired object classes may not exist in the dataset. The ideal dataset should contain images representing objects of a suitable size, and the amount of data should be evenly distributed for all classes.

#### 2.3.3. Number of Anchor Boxes

Most object detection methods use the region proposal method. In this method, an anchor box is an element that is defined for the region to be considered in the extracted feature map and is used to train whether an object in the box exists. As the number of anchor boxes increases, an accurate bounding box can be obtained because the object’s existence area is detected at various scales from the feature map. Fewer resources are required to process anchor boxes than those required by the backbone layer, but excessively increasing the number of anchor boxes can affect the overall runtime. There is a trade-off relationship between the detection accuracy and the overhead of post-processing, such as non-maximum suppression (NMS).

### 2.4. Lightweight Network

Because high-performance computational processing units are concentrated in a data center environment, any DNN model can perform quickly when provided with high computational capabilities. However, there is a limitation in performing complex DNN models on devices in a single-CPU or single-GPU environment.

#### 2.4.1. MobileNet

MobileNet is a lightweight network for object detection in resource-constrained environments, such as mobile devices and single embedded boards [20]. To design a shallow network, MobileNet does not contain a fully connected layer and has a max-pooling layer on the high-level resolution layers to quickly downsample the dimensions. In addition, the computational efficiency is increased by a factor of eight compared to the convolution layer with depth-wise separable convolution using a 1 × 1 kernel.

#### 2.4.2. YOLO-LITE, YOLO Nano and Tiny-YOLO

The YOLOv3 model has 68 million training parameters with a total of 106 layers, including three prediction layers [21]. Although YOLOv3 can detect 80 classes with high accuracy, it is difficult to perform real-time object detection in a non-GPU environment. YOLO-LITE, YOLO Nano and Tiny-YOLO are simplified models of YOLO for object detection in a non-GPU environment [22,23,24]. YOLO-LITE is composed of seven convolution layers and trains a model based on five regions. It is composed of a small number of layers and performs object detection quickly, but its accuracy performance is poor. YOLO Nano is highly compact DNN model for the task of object detection. To reduce operation cost, YOLO Nano contains module-level macro architecture and micro architecture designs tailored for the task of embedded DNN inference. Alternatively, Tiny-YOLO consists of 23 layers, including two prediction layers (YOLO), and detects objects using six anchor boxes. Figure 2 shows the Tiny-YOLO network structure. Tiny-YOLO reduces the dimensionality quickly, similar to MobileNet, and reduces the size of the backbone network by using a narrow channel. Tiny-YOLO can quickly detect objects in a non-GPU environment through its 6.8 million learning parameters.

## 3. Domain-Specific Object Detection

In this section, we introduce the definition of OOI groups for domain-specific object detection and describe how to solve the class imbalance problem of open datasets. We also discuss various network structures for shallow networks and propose a lightweight model structure.

### 3.1. Domain-Specific OOI Groups

Existing object detection models are trained for many classes and can be applied to a variety of domains. For example, 80 classes can be detected for models trained with the MS COCO dataset and 100 classes for models trained with the CIFAR-100 dataset [17,25]. However, classes in a dataset cannot be highly relevant to all domains. For example, vehicle-type objects are highly relevant to highway domains, and objects such as people, trees, and benches are highly relevant to park domains. That is, the range of objects to be detected is limited according to the domain. Figure 3 illustrates how to define the OOI group for each domain. Object detection is performed using the SOTA DNN model on randomly extracted data from videos for each domain, and the detection frequencies for each object are calculated. Objects detected with high frequency are highly domain-relevant objects, thus we define these objects as the OOI group.

We generate a training dataset after defining domain-specific OOI groups. Open datasets, such as MS COCO and PASCAL VOC, provide vast amounts of data for various object classes. However, because these open datasets cannot include all classes, there may be differences in detection accuracy because of data sparsity problems. In addition, even within the same class, training data with low relevance to the domain may exist. For example, cars exhibited at motor shows have low relevance to the highway domain, and we should remove those images from the dataset. However, checking and filtering all the data is time-consuming. For classes with insufficient datasets, we should generate training data from the sampled data.

Figure 4 shows an example of the generation of training data for OOI groups. If ships, cars, and trucks are detected, the location and information of the car and truck classes are stored, and the ship is excluded as it is not in the OOI group. In this study, training dataset are generated using continuous frame extraction based on YouTube videos, allowing more data to be generated at different sizes and angles for the same object.

### 3.2. Lightweight Object Detection Model Structure

For on-device object detection, the domain-specific model should consist of few training parameters. We conducted an empirical search for an optimal model design. Figure 5 shows the baseline model structure used for the empirical search. The baseline structure was a mixed structure of Tiny-YOLO and YOLOv3. To design the shallow and narrow network, two convolution layers with a stride value of two were used to quickly reduce the dimension, and the maximum number of filters was set to 512. The baseline model size is 15.1 MB, which is approximately 94% lighter than the YOLOv3 model and approximately 44% lighter than the Tiny-YOLO model. The baseline model requires less than 10 min to train using 18,000 images on a desktop equipped with an RTX 2060 SUPER GPU.

#### 3.2.1. Partial Routing for Feature Reuse

Since the lightweight network consists of a small number of layers, it has fewer training parameters to perform feature extraction. To perform high-accuracy object detection with a small number of training parameters, it is necessary to reuse the features extracted from each layer. Figure 6 shows the designed partial routing structure for feature reuse. All layers share feature maps with other layers with the same dimensionality; the feature maps are merged at each prediction layer by concatenating operations in three branches. The model performance can be improved using the partial routing structure because the concatenate operation is performed to maintain the unique feature map. The size of the network with the partial routing structure added is 22.8 MB, which is approximately 33% lower than that the Tiny-YOLO model.

#### 3.2.2. Dual-Residual Block

Since the shallow network has fewer learning parameters, it has limitations in extracting precise feature maps. The baseline model is a structure in which a convolutional layer and a max-pooling layer are repeated, and the dimensionality is rapidly reduced to a low-level resolution. Additional layers should be placed to extract feature maps of the high-level resolution of the baseline model. We design a backbone network, as depicted in Figure 7, to extract feature maps of high-level resolutions while minimizing the model complexity. Figure 7a is the residual block used in ResNet, and Figure 7b is a dual-residual block, the extension of the residual block, which allows the extraction of feature maps for multiple dimensions by using max-pooling and upsampling layers simultaneously. The dual-residual block performs a concatenate operation to reuse the unique feature maps of different channels. The size of the model using the residual block is 15.5 MB, which is approximately 55% lower than that of the Tiny-YOLO model, and the size of the backbone with the dual-residual block is 20.6 MB, which is approximately 55% lower than that of Tiny-YOLO.

#### 3.2.3. Prediction Layer

The prediction layer converts the dimensions of the extracted feature map into the output dimensions. The number of channels in the output layers depends on the number of anchor boxes and classes. Because the number of classes is dependent on the domain, as the number of classes increases, additional layers should be placed to ensure the accuracy of the detection model. Figure 8 shows the structure of the OOI group’s adaptive prediction layer. To maintain the shallow network, we reduce the number of channels by half until the number of channels is less than 32 times the number of classes. When there are five classes in the OOI group, the model size is 23.2 MB.

#### 3.2.4. Proposed Lightweight Object Detection Model

Figure 9 shows the lightweight network structure with the best performance as measured by an empirical search for on-device object detection. The details of the experiment are described in Section 5. With a structure in which the dual-residual block and OOI group adaptive prediction layer are added to the baseline model, the model size is 25.8 MB, which is approximately 36% smaller than the Tiny-YOLO model.

## 4. Experiments and Discussion

### 4.1. Experimental Environment Setup

To evaluate the performance of the domain-specific on-device object detection method proposed in this study, we conducted a comparative experiment with YOLOv3-SPP, Tiny-YOLO, and YOLO Nano. YOLOv3-SPP is a version of YOLOv3 that contains a spatial pyramid pooling (SPP) block. The training was performed on a desktop computer equipped with an Intel Core i7 10700K CPU, 16 GB RAM, and an NVIDIA RTX 2060 SUPER GPU. All models were implemented using the PyTorch framework. GPU operations were performed in an environment where CUDA 10.1 version and cuDNN 8.1 version software were installed.

#### 4.1.1. Experimental Dataset

In this study, we limited the domains to streets and parks for our experiments. To define the OOI groups, six YouTube videos for each domain were used, and the DETR model was used as the SOTA DNN model [26]. Training datasets were generated using MS COCO 2017, Open Image v4, and randomly sampled images [17,27]. We defined OOI groups from the street and park domains as follows: car, bus, motorcycle, traffic light, stop sign, truck, and person, bench, bicycle, umbrella, ball, and dog, respectively. Table 1 shows the number of images and the number of classes used for training each domain.

#### 4.1.2. Evaluation Metrics

We used three metrics for performance evaluation, precision, recall, and f-measure, as given in Equations (1)–(3) [28]. True positive (TP) refers to a case in which an OOI object is detected. The higher the number of TPs, the higher the accuracy of the model. False positive (FP) refers to when OOI objects are detected but incorrectly classified, and false negative (FN) refers to when the OOI object is not detected. Precision is the proportion of correctly detected objects among true answers. Recall denotes the proportion of the true answers among detected objects, that is, recall refers to the hit ratio of the object detection results. The f-measure is the arithmetic mean of precision and recall.
(1)$$Precision=\frac{TP}{TP+FP}$$
(2)$$Recall=\frac{TP}{TP+FN}$$
(3)$$f-measure=\frac{2*Precision*Recall}{Precision+Recall}$$

### 4.2. Experimental Results for Different Trials

Various object detection models, from Trials 1 to 7, were defined and evaluated to find the optimal lightweight model, as shown in Table 2. For all trials, 90 epochs of training were carried out using the MS COCO 2017 dataset, Open Image v4, and the randomly sampled image training dataset to obtain the optimal model. The input image size used in the model training and testing was set to 416 × 416, which is consistent with that of the Tiny-YOLO model. For fairness of the experiment, the YOLOv3-SPP, Tiny-YOLO and YOLO Nano models were trained for 90 epochs and tested in the same experimental environment and parameter settings. The experimental results for all the trials are shown in Table 3. The test dataset contained 1337 images of Open Image v4, MS COCO 2017, and randomly sampled images. The details of the experiment are as follows.

#### 4.2.1. Deep Network and Shallow Network

The YOLOv3-SPP model has the most training parameters of all networks. Theoretically, the YOLOv3-SPP model should perform best in terms of precision and recall metrics. However, we observed that the YOLOv3-SPP model had the lowest precision performance. The YOLOv3-SPP model overfits the training data because it has too many training parameters for a small number of classes. The Tiny-YOLO and YOLO Nano model were 23.5% and 11.8% better than YOLOv3-SPP in the precision metric, respectively. In addition, in an environment with limited resources (i.e., only a CPU), the object detection speed of the Tiny-YOLO model was approximately 10 times faster than that of the YOLOv3 model. Theoretically, YOLO Nano model should have the fastest detection speed, but Tiny-YOLO and the Baseline model are more faster detection rates than YOLO Nano. Although YOLO Nano has 1 × 1 and 3 × 3 size kernels of narrow channels between layers, it includes 100 convolution blocks. Therefore, YOLO Nano has a deeper network than Tiny-YOLO and baseline models. We confirm that a shallow network is more suitable than a deep network as a backbone network structure for on-device object detection.

#### 4.2.2. Partial Routing for Feature Map Reuse

Because the shallow network consists of a small number of layers, the feature map that can be extracted is smaller than that of a deep network. To solve this problem, the partial routing structure shown in Figure 6 was applied to the baseline model (Trial 1). The precision of Trial 1 was increased by 1.8%, but the object detection speed was decreased by 3.3 FPS in the CPU environment. Comparing Trial 1 with a baseline model, the precision increased slightly, but the number of parameters and model size increased remarkably. We confirm that the partial routing structure for feature reuse is not suitable for domain-specific on-device object detection.

#### 4.2.3. Residual Block and Dual-Residual Block

To extract feature maps of the high-level resolution, Trials 2 and 3 added residual blocks and dual-residual blocks, respectively, to the backbone network. Compared with the baseline model, the precisions of Trials 2 and 3 were improved by 4.2% and 9.3%, respectively, and for Trial 3, recall was significantly improved by 14.1%. Although the detection speed of Trial 3 increased by approximately 24% compared with the baseline model, it was fast enough to perform object detection in the CPU environment. These results show that the residual block and the dual-residual block are suitable for lightweight networks. In addition, when dual-residual blocks were used instead of the residual blocks, the precision and recall were improved by 5.1% and 9.1%, respectively. This confirms the efficiency of the dual-residual block in shallow networks.

#### 4.2.4. OOI Adaptive Layer

Trials 5, 6, and 7 used the OOI group adaptive prediction layers to extract additional feature maps of the output layers. Compared with the baseline model, the precision and recall of Trial 5 were improved by 6% and 10%, respectively. For Trial 6, the precision and recall increased by 10.8% and 14.8%, respectively, compared with the baseline model. Thus, we confirmed that the most influential element for the performance improvement of the lightweight model is to use the dual-resident block and the OOI group adaptive prediction layer simultaneously (Trial 6).

### 4.3. Experimental Results for Different Dataset

#### 4.3.1. MS COCO 2017 Dataset

To evaluate the detailed performance of Trial 6, which is the best lightweight model, we used the MS COCO 2017 dataset. Table 4 shows the precision and recall measurements for the street domain classes. The precision and recall were measured up to 82.26% and 77.69% for cars, buses, trucks, and motorcycles, respectively, for which data collection is relatively easy. However, there was still a class imbalance problem for stop signs and traffic light classes, which are relatively difficult data to obtain. We found that in deep networks, such as the YOLOv3-SPP model, the problem of data imbalance did not significantly affect the performance degradation, whereas the problem of data sparsity in the shallow network had a significant impact on performance degradation. A portion of the experimental results for the Trial 6 model using the MS COCO 2017 dataset are shown in Figure 10.

#### 4.3.2. PASCAL VOC 2007 and Open Image v4

To evaluate the detection performance of the Trial 6 model on data that contained objects of different scales, we conducted experiments on the PASCAL VOC 2007 and Open Image v4 datasets. Object classes such as traffic lights, stop sign, truck, bench, tree, um-brella, and ball, which did not exist in the PASCAL VOC 2007 dataset, were collected from the Open Image v4 dataset; the test dataset contained 938 images from both datasets. To evaluate the general applicability of the proposed lightweight model to object detection tasks, an experiment was conducted without any additional model training. Table 5 shows the precision and recall results for the person, dog, bicycle, bus, car, and motorcycle classes included in the PASCAL VOC 2007 and Open Image v4 datasets. The precision of the car, traffic light, stop sign, and truck classes were revealed to be 5.16, 10.46, 5.35, and 2.67% higher than when tested with the MS COCO dataset, respectively. However, the precision of the bus and motorcycle classes decreased by 7.91 and 1.93%, respectively. These results confirmed the applicability of the proposed lightweight model to object detection in various datasets. The partial test results of the Trial 6 model for the PASCAL VOC 2007 dataset are shown in Figure 11.

## 5. Conclusions

In this paper, we proposed a domain-specific on-device object detection method. To perform domain-specific object detection, we defined objects with a high frequency of detection as OOI groups by performing object detection based on the DETR model with images sampled from domain-related videos. The training data, which contained objects of the OOI group, consisted of the MS COCO 2017 dataset, Open Image v4, and randomly sampled data, and were used to train the domain-specific detection model. To design the shallow network structure for each domain, high-level resolution features were extracted at minimal cost using dual-residual blocks. To consider the OOI group characteristics, we used the OOI group adaptive prediction layers. The experimental results reveal that the size of the proposed lightweight model is 21.7 MB, which is 91.35% and 36.98% smaller than that of YOLOv3-SPP and Tiny-YOLO, respectively. The precision scores achieved using the MS COCO 2017 dataset were 9.9% and 33.4% higher than those of Tiny-YOLO and YOLOv3-SPP, respectively. In addition, the proposed lightweight model performed object detection approximately 7.5 times faster than the YOLO-SPPv3 model in a CPU-only environment. These results prove that the proposed lightweight model can achieve higher efficiency and better performance than existing models on non-GPU devices, such as mobile devices and embedded boards. The proposed object detection method satisfies the latency and accuracy requirements; thus, it can process continuous data such as video in real time. However, because one detection model per domain is required, there is a problem that the overhead of preprocessing such as OOI group definition and model training is too large to apply it to various domains. In future work, we aim to design a generalized model applicable to multiple domains with an attention-based network structure.

## Figures and Tables

**Figure 1 entropy-24-00077-f001:**
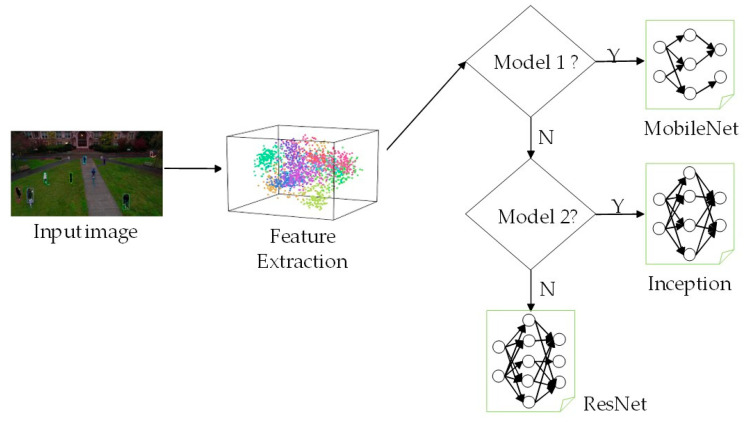
Example of the model selection.

**Figure 2 entropy-24-00077-f002:**
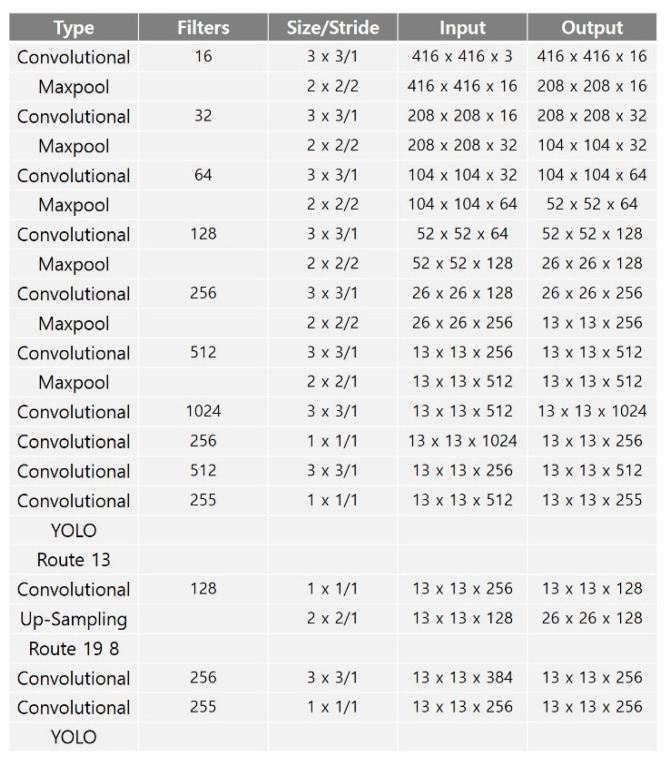
Tiny-YOLO network structure.

**Figure 3 entropy-24-00077-f003:**
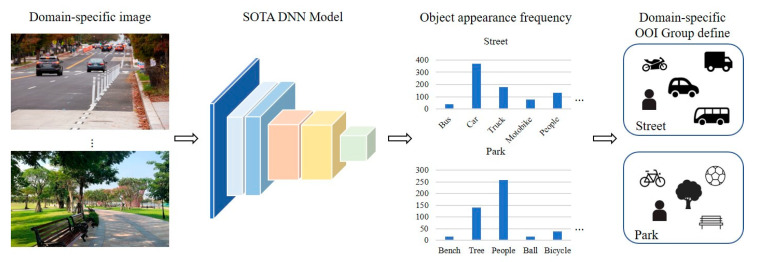
Decision process of domain-specific OOI groups.

**Figure 4 entropy-24-00077-f004:**
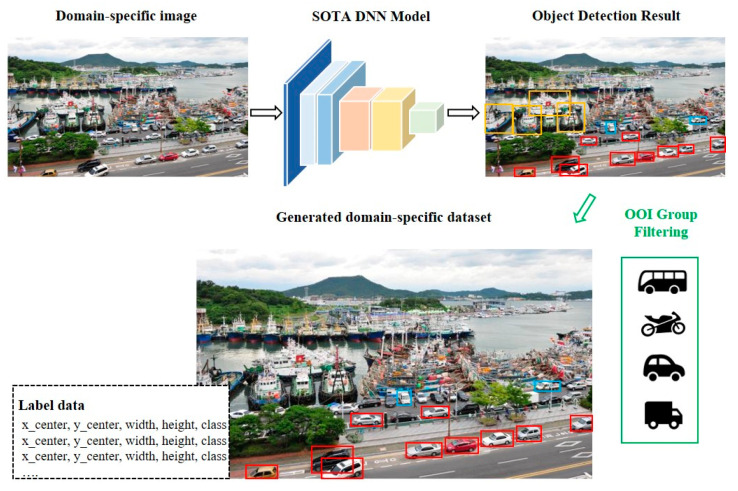
Example of the generation of training data for OOI groups.

**Figure 5 entropy-24-00077-f005:**
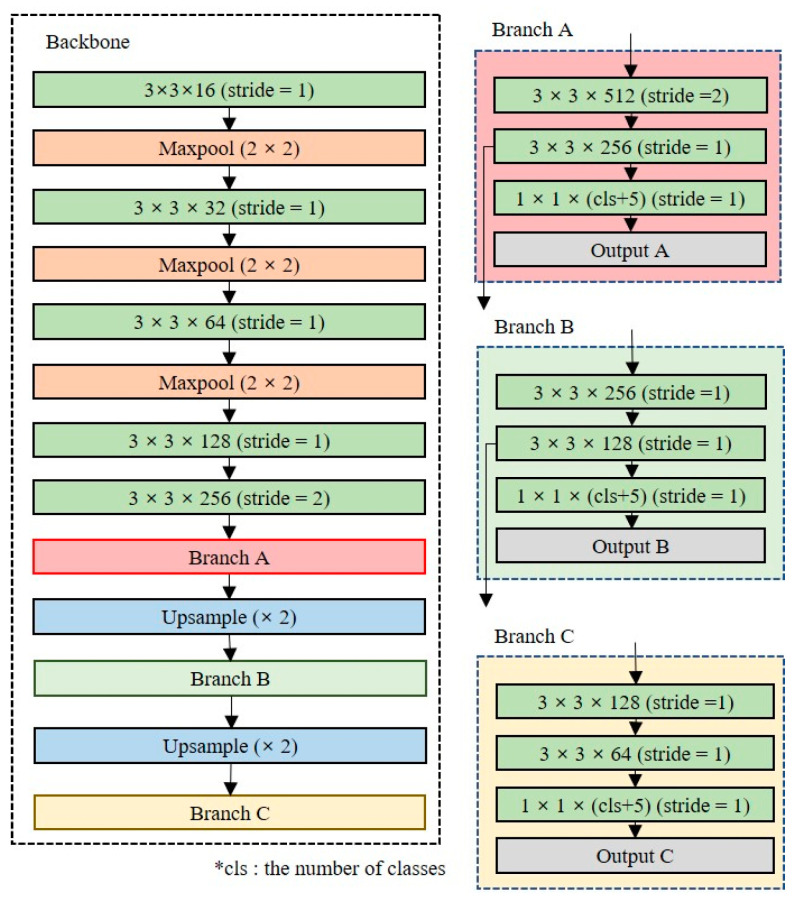
Baseline network structure, which is a mixed structure of Tiny-YOLO and YOLOv3.

**Figure 6 entropy-24-00077-f006:**
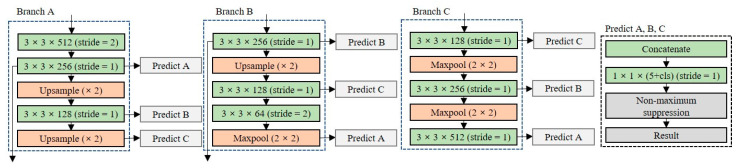
Partial routing structure for feature reuse.

**Figure 7 entropy-24-00077-f007:**
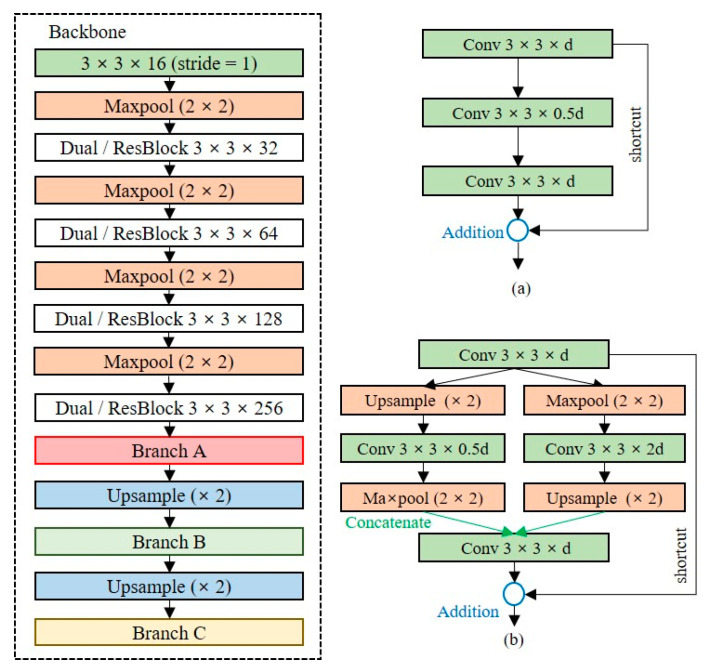
Shallow network structure for high-level resolution feature extraction, (**a**) residual block, (**b**) dual-residual block.

**Figure 8 entropy-24-00077-f008:**
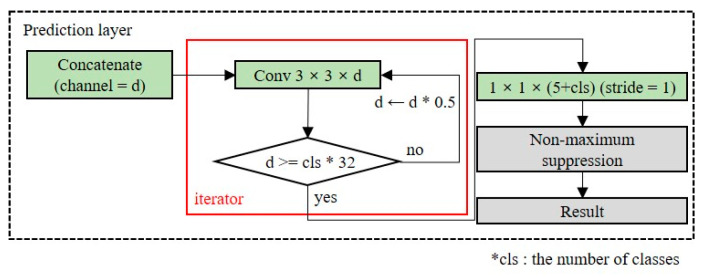
Structure of the object of interest (OOI) group adaptive prediction layer.

**Figure 9 entropy-24-00077-f009:**
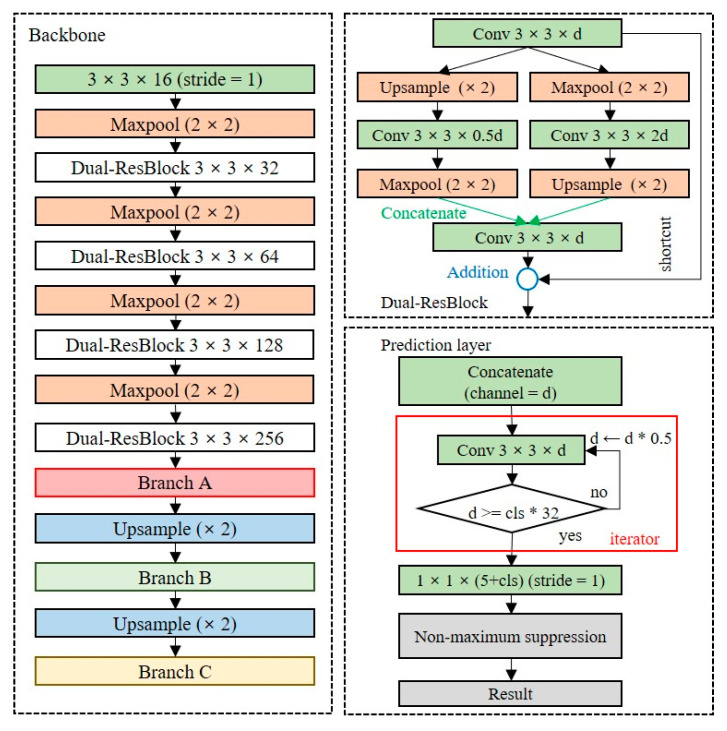
Proposed lightweight network structure for on-device object detection.

**Figure 10 entropy-24-00077-f010:**
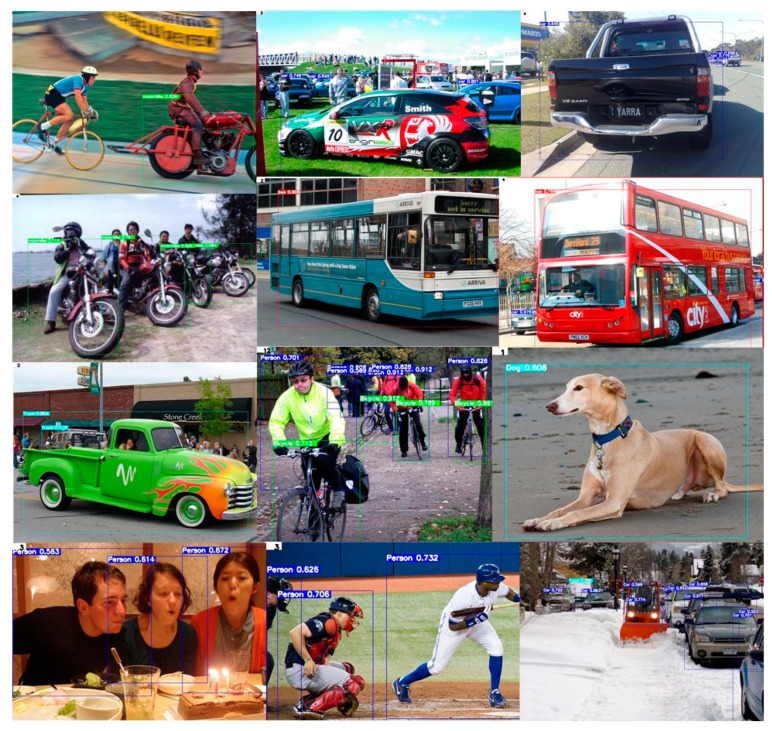
Partial test results of the Trial 6 model for MS COCO 2017 dataset.

**Figure 11 entropy-24-00077-f011:**
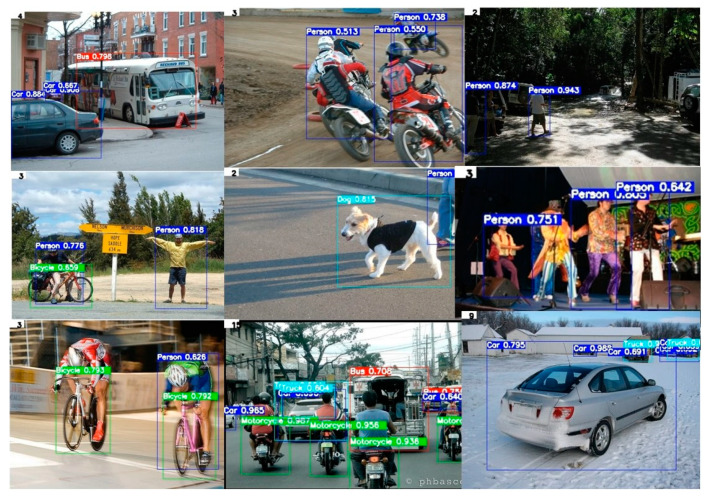
Partial test results of the Trial 6 model for PASCAL VOC 2007 dataset.

**Table 1 entropy-24-00077-t001:** Overview of number of images and number of classes for each domain.

Domain	Training Dataset	Validation Dataset	Number of Classes
Street	29,073	1252	6
Park	18,350	899	6

**Table 2 entropy-24-00077-t002:** Network structure descriptions of different trials.

Model	Structure Description
Baseline	Backbone uses Tiny-YOLO while using yolov3-spp detector as the branch (Figure 5)
Trial 1	Based on baseline, using partial routing structure at the branch layer (backbone of baseline + Figure 6)
Trial 2	Based on baseline, using a residual block after the max-pooling layers (baseline + Figure 7a)
Trial 3	Based on baseline, using a dual-residual block after the max-pooling layers (baseline + Figure 7b)
Trial 4	Based on Trial 1, using a dual-residual block after the max-pooling layers (backbone of baseline + Figure 6 + Figure 7b)
Trial 5	Based on baseline, using an OOI adaptive prediction layer every branch layer (backbone of baseline + Figure 8)
Trial 6	Based on Trial 3, using an OOI adaptive prediction layer every branch layer (baseline + Figure 7b + Figure 8)
Trial 7	Based on Trial 4, using an OOI adaptive prediction layer every prediction layer (backbone of baseline + Figure 6 + Figure 7b + Figure 8)

**Table 3 entropy-24-00077-t003:** Results of the different trials. The best performances are labelled out in bold.

Model	Size (MB)	Precision	Recall	F-Measure	FPS (GPU)	FPS (CPU)
YOLOv3-SPP	224.8	0.418	**0.934**	0.577	20.67	1.26
Tiny-YOLO	33.9	0.653	0.630	0.641	64.45	12.65
YOLO Nano	11.5	0.536	0.599	0.557	51.20	9.12
Baseline	**15.1**	0.644	0.622	0.632	**67.12**	**14.30**
Trial 1	22.8	0.666	0.605	0.634	61.15	11.00
Trial 2	15.6	0.686	0.672	0.679	66.61	12.72
Trial 3	20.6	0.737	0.763	0.749	59.02	10.91
Trial 4	23.3	0.717	0.702	0.709	58.13	9.57
Trial 5	15.3	0.704	0.722	0.713	61.41	14.05
Trial 6	21.7	**0.752**	0.770	**0.760**	57.70	9.42
Trial 7	25.8	0.720	0.699	0.709	54.40	8.30

**Table 4 entropy-24-00077-t004:** Evaluation results for Trial 6 using MS COCO 2017 dataset.

**Class Name**	**Car**	**Bus**	**Truck**	**Stop Sign**	**Traffic Light**	**Motorcycle**
Precision (%)	64.39	82.26	58.51	53.35	45.66	69.32
Recall (%)	62.18	77.96	74.47	48.76	47.12	65.15
**Class Name**	**Person**	**Bench**	**Bicycle**	**Umbrella**	**Ball**	**Dog**
Precision (%)	62.28	61.33	62.23	49.12	68.16	68.71
Recall (%)	60.65	63.90	60.57	53.06	71.40	67.74

**Table 5 entropy-24-00077-t005:** Evaluation results for Trial 6 using PASCAL VOC 2007 and Open Image v4 dataset.

**Class Name**	**Car**	**Bus**	**Truck**	**Stop Sign**	**Traffic Light**	**Motorcycle**
Precision (%)	69.55	74.35	61.18	58.70	56.12	67.30
Recall (%)	71.84	68.24	55.50	51.47	48.90	58.61
**Class Name**	**Person**	**Bench**	**Bicycle**	**Umbrella**	**Ball**	**Dog**
Precision (%)	70.90	67.51	62.46	51.33	68.92	66.24
Recall (%)	68.02	54.53	59.08	53.04	75.20	57.98

## Data Availability

Not applicable.

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
