# Peer review of "Domain-Specific On-Device Object Detection Method"

_entropy, 2022, doi:10.3390/e24010077_

Round 1

Reviewer 1 Report

The manuscript has improved since its last version, with clearer Figures and more detailed explanations. The authors' replies to my comments are satisfactory.

Reviewer 2 Report

This paper proposed a domain-specific on-device object detection method, this is a lightweight network. However, there is a problem of high pre-processing overhead. In Figure 6, the difference between upsample(2X) and upsample(X2) should be given. The figures and tables are suggested to be updated for better appearance and easier reading. For example, the best performance could be labelled out in bold in Table3. 

Reviewer 3 Report

This work proposes a domain-specific on-device object-detection method, which defines objects with a high frequency of appearance as the object of interest (OOI) group . The detection model can detect objects using a shallow network structure and ensures that latency requirement is met by reducing the complexity of the detection models. This method is interesting for real-word applications, and is suitable for on-device environments. The experimental results demonstrate the performance. The reviewer's main concern is about the baseline methods. YOLOv5 is the updated detection method, why does it not been compared?

Author Response

This manuscript is a resubmission of an earlier submission. The following is a list of the peer review reports and author responses from that submission.

Round 1

Reviewer 1 Report

In this paper, to address the issues in lightweight network for object detection, the authors proposed how to define the OOI group for each domain and how to generate training datasets for each OOI group, and presented several structures in the experiments.

The idea sounds good, and the experimental results on the datasets are fine. However, there are some problems need to be solved:

  1. For the writing.

There are some grammatical errors. Please carefully check and correct them.

Some graphs is not clear enough, Such as:

‘Table 2. Network structure descriptions for each trial.’ The network structure described is quite unintuitive in comparison.

Figure 5, 7, 8 and 9, it is hard to read such horizontal layout network structure.

Ⅱ. For the keywords.

The number of keywords is too much, which seems to be not focused enough. Please try to reduce the number of keywords.

III. For the Introduction and related works.

It is unnecessary to introduce the development of computer vision from the beginning, or even introduce image recognition. Please focusing on the introduction of object detection.

The content ‘Previous Research in Resource-constrained Environment’ is relatively inadequate.

Some of the statements in “Introduction” are not accurate and too broad. E.g. “However, these approaches can lead to poor performance in accuracy-critical application” How poor performance they are? What is the accuracy-critical application?

  1. For the experiments

The experiments are not enough. The article only compares the lightweight model: Tiny-YOLO. As far as I know, more lightweight and faster models appear in the field of object detection, such as Yolo-nano, Micro-YOLO. More evaluation indicators are needed.

About the optimal trail in the paper, please supplement the experiment and explain the reason for the optimal trail.

  1. For the idea of OOI

“… using the SOTA DNN model on randomly extracted data from videos for each domain, and the detection frequencies for each object are calculated.”

It is difficult to distinguish the definitions of different domains strictly. The street and park used in the article are of particularity.

I think it is better to show the efficiency in this decision-making process.

Please report the development potential of this domain specifical object detection method.

  1. Some important references are missed, e.g:

[R1] Wu, J., Leng, C., Wang, Y., Hu, Q., and Cheng, J. (2016). Quantized convolutional neural networks for mobile devices. In IEEE Conference on Computer Vision and Pattern Recognition, 2016, pp. 4820-4828.

[R2] Z. Tu, Z. Guo, W. Xie, M. Yan, R. Veltkamp, B. Li, and J. Yuan. Fusing disparate object signatures for salient object detection in video. Pattern Recognition, vol.72, pp.285–299, 2017.

[R3] Rastegari, M., Ordonez, V., Redmon, J., and Farhadi, A. Xnor-net: Imagenet classification using binary convolutional neural networks. In European conference on computer vision, 2016, pp. 525-542.

Reviewer 2 Report

This paper uses a deep neural network (DNN) models for domain-specific on-device object-detection on the basis of preselected object of interest (OOI) groups containing high frequency data in specific domains. Compared with existing DNN models, the layers of the domain-specific model are resricted and the number of trainable parameters reduced to speed up object detection. Various network structures were combined to obtain the best-performing "lightweight detection" model. Detection precision and recall were found to outperform other models. The authors state that their lightweight model performs significantly better than others on mobile devices and embedded boards. 

The paper addresses a particular space of interest and has the merit of narrowing down the parameter space of the DNN model approach in the context of a domain specific application to obtain a less heavy and at the same time more accurate detection performance. However, DNNs are not the only solution to the problem(s) addressed in this work here, and some important reference work has not been taken into consideration to justify why a DNN based approach would be the first choice model in the first place.

1. Automated video object recognition is a topic of emerging importance in both defense and civilian applications. Accurate and low-power neuromorphic architecture systems for real-time automated video object recognition have been proposed earlier by others. Neuormorphic visual understanding of scenes is inspired by computational neuroscience models of feed-forward object detection and a prime chocie for processing visual data, enabling object detection based on form and motion modeling, and object classification based on convolutional neural networks. Such models were successfully evaluated on urban area video datasets collected from a mix of stationary and moving platformsis, and their performance is potentially superior to that of the DNN method described here. Neuromorphic model architecture with high object recognition accuracy and minimal energy consumption for a large number of objects of different types in cluttered scenes with varying illumination and occlusion conditions have been developed earlier by others. These approaches should be discussed in the context of this study to clarify both the specific interest, and the limitations of the domain-specific DNN lightweight model developed here.

2. How does the proposed approach solve the problem of moving object detection? Manipulating and accurately merging information coming from subsequent video frames while deploying minmal computational efforts to each single frame is a current challenge here. Combining several known and already tested models represents an interesting solution for obtaining optimal detection results in diversified scenarios where objects can and will change positions. The authors should place and discuss their region of interest approach within the  larger scientific context of automatic detection methods deployed to efficiently track object movement. For example, how would the DNN detect and/or correct false positives in this case? Accuracy and reliability mainly depend on the overall adequacy of the software system designed and implemented, i.e. how the different algorithmic phases communicate and collaborate with each other to concur toward the best, domain-specific solution.  Discussion of such perspectives is missing from this paper. The conclusions and perspectives section should be revised to clarify 1) the real perspectives and 2) the many limitations of the DNN model developed here.
